# Predicting the Future Distribution of *Leucobryum aduncum* under Climate Change

Puwadol Chawengkul [1], Patsakorn Tiwutanon [1], Nuttha Sanevas [1] and Ekaphan Kraichak [1,2,*]

1   Department of Botany, Faculty of Science, Kasetsart University, Bangkok 10900, Thailand;
    puwadol.c@ku.th (P.C.); patsakorn.a@ku.th (P.T.); nuttha.s@ku.th (N.S.)
2   Biodiversity Center, Kasetsart University, Bangkok 10900, Thailand
*   Correspondence: ekaphan.k@ku.th

**Abstract:** *Leucobryum aduncum* is a moss species reported in many Southeast Asian regions, often found in forests with a high humidity. Climate change may impact the future distribution of this species. This study aimed to model the current distribution and predict the impact of climate change on *L. aduncum* distribution in the next 50 years across Southeast Asia. In the process, relevant climate variables in the distribution of the species were also identified. The occurrence data of this species with current and future climate models from CMIP6 under moderate (SSP2) scenarios were used to predict current and future *L. aduncum* distributions. Under the current climate, the predicted suitable areas for *L. aduncum* included most mountainous areas. However, many Southeast Asian areas showed a lower probability of finding this species in the next 50 years. The distribution area of this species will dramatically decrease by 50.16% in the current area. The most important ecological variables included the "mean temperature of the driest quarter" and the "annual temperature range". This study suggests the possible impacts of an increased temperature and the scale of climate change on the distribution of sensitive plants like bryophytes.

**Keywords:** bryophytes; global warming; species distribution modeling

## 1. Introduction

Bryophytes are non-vascular plants that usually grow in an environment with high moisture [1]. In the forest ecosystem, they assist in storing a large amount of water, which plays an important role in the water balance and helps to reduce erosion along streams and landslides. They grow in high areas above sea level and are the biological indicators of an abundant environment with high moisture. Furthermore, they provide microhabitats for small animals, insects, and microorganisms, food for some beetles, and a seedling bed for germination [2,3]. Bryophytes are also pioneer plants that grow in high areas above sea level, flooded areas, and even wastelands [4]. After bryophytes die, they accumulate into humus soil that promotes other land plants. Furthermore, they also reduce erosion along streams and remove heavy metal ions from the wastewater [5].

The distribution of bryophytes has been particularly sensitive to the climate, especially temperature and precipitation. The success of colonization relies more on suitable ecological conditions than the dispersal process [6]. The seasonal distribution of rainfall and temperatures are key to maintaining the diversity of bryophyte communities [7]. In Southeast Asia, water and precipitation played a dominant role in describing the variation of bryophyte species diversity, while woody species were much more strongly affected by temperature [8]. The interaction between water and energy might play an important role. Furthermore, the richness of the sampling scale is mainly affected by the richness of the local area, which is also controlled by the climate [9].

*Leucobryum* is a unique group of bryophytes, forming cushions in moist habitats, such as on soil, bark, and even wet rock. The plants are whitish-green when moist and become

whitish when dry. Fourteen species of *Leucobryum* were recorded in many areas around Southeast Asia [10], indicating the diversity hotspot of *Leucobryum* and the abundance of the forest in Southeast Asia. However, a unified database on the species occurrence of *Leucobryum* in Southeast Asia is still unavailable. Many areas in Southeast Asia lack surveys but may have a high potential to find these species, and still require to focus more on area-based research on species diversity [11].

*Leucobryum aduncum* is one of the most common species of *Leucobryum* in Southeast Asia. It is widely distributed from southern China to New Guinea (Figure 1a). This species is often found growing in the shade of trees in areas with high moisture, such as moist rocks under forests or locations with high rainfall [12–14]. Nowadays, *L. aduncum* is at a high risk of population decline due to both the direct and indirect activities of humans. Because of its beautiful appearance and incredible ability to absorb water, it is heavily harvested from the forest for various commercial purposes, including as a potting medium, soil cover, and for terrarium use. In many groups of mosses, climate change may impact the population size [15–20] and genetic diversity [21,22] due to the anthropogenic warming of the climate. However, such effects have not been studied in *Leucobryum*.

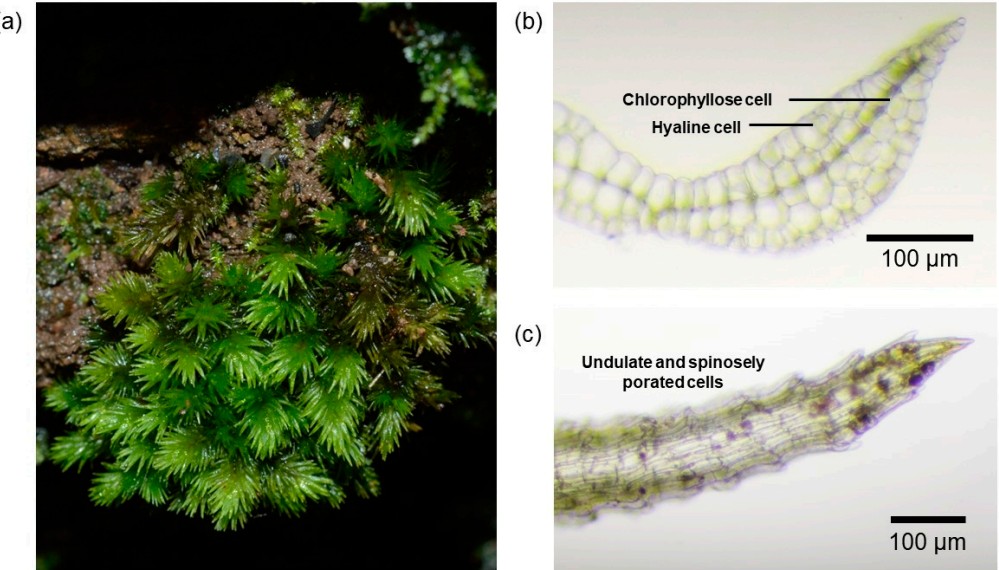

**Figure 1.** Habit and anatomical structure of *Leucobryum aduncum*: (**a**) Habit of *Tiwutanon 53* Hb. Kasetsart University; (**b**) cross-section of leaf showing hyaline cells and chlorophyllose cells, and (**c**) the projecting end of undulate and spinosely porated cells.

Species distribution modeling (SDM) is a tool for predicting the potential distributions of any living organisms and the relationship between the occurrences of the species and ecological variables. The results of the model will present the probability of finding the species in each area. SDM has been widely used in previous studies across various types of areas (such as terrestrial, freshwater, and marine), including the study of bryophyte species distribution [23,24].

*Leucobryum aduncum* provides an intriguing case study for predicting responses to climate change, because of its unique morphological and ecological characteristics. *L. aduncum* can be found in various habitats, from beach forests to montane forests [14,25], suggesting a broad tolerance range. Unlike typical bryophytes, this species thrives in a relatively arid environment, owing to its capacity to retain extra water within its water-storage cells (Figure 1b) and between plants in a cushion [26]. Additionally, the leaf apex features undulate and spinosely porated cells that help to improve water-intake efficiency [14] (Figure 1c). Identifying the relevant climatic variables for *L. aduncum* distribution will deepen our understanding of the physiological and ecological requirements for this species. The results will help us to discover possible habitats to find *L. aduncum*, and to plan for the

new survey to fill our knowledge gaps. Furthermore, this tool can also predict the potential distributions of *L. aduncum* using the ecological variables related to future climatic data. The distributions of these species may change in the future due to the effect of climate change.

This study aimed to predict the impact of climate change on *L. aduncum* distribution in the next 50 years and to determine the ecological variables related to the species distribution of this species in Southeast Asia. The current work employed the Maxent algorithm with selected bioclimatic variables to predict the current distribution. Future climate variables from six possible CMIP6 scenarios were used to predict the future distribution. The results from the study would provide us with insights into the impact of climate change and possible conservation plans for this understudied group of plants.

## 2. Materials and Methods

### 2.1. Data Acquisition

The workflow of species distribution modeling in this study is outlined in Figure 2. The occurrences of *Leucobryum aduncum* distribution in Southeast Asia were downloaded from the Global Biodiversity Information Facility (GBIF: https://www.gbif.org/, accessed on 3 October 2022) and extracted from herbarium specimen labels deposited in the herbarium of the Department of Botany at the Kasetsart University, Bangkok Forest (BKF), Burapha University, and Prince of Songkla University (PSU) Herbaria. Southeast Asia was defined by the area between the latitude of 10° S and 23° N and the latitude of 95° E and 140° E. *Leucobryum aduncum* var. *scalare* was excluded from the analysis due to its reclassification as a separate species [14]. The occurrence data were pre-processed via converting latitude and longitude coordinates from a degrees/minutes/seconds (DMS) to a decimal degrees (DDs) format using Microsoft Excel version 2307. Finally, 172 occurrences were exported as a .csv file for generating a species distribution model. Due to the limitation in sampling size, all occurrences were used for modeling. However, to reduce the risk of spatial autocorrelation, the occurrences were split into four equally sized spatial blocks for the subsequent analyses.

The 19 bioclimatic variables were retrieved from WorldClim version 2.1 [27] at a resolution of 2.5 arc-minutes (from https://www.worldclim.org/, accessed on 3 October 2022). The variables included the annual mean temperature (bio1), the mean diurnal range (bio2), the isothermality (bio3), the temperature seasonality (bio4), the maximum temperature of the warmest month (bio5), the minimum temperature of the coldest month (bio6), the annual temperature range (bio7), the mean temperature of the wettest quarter (bio8), the mean temperature of the driest quarter (bio9), the mean temperature of the warmest quarter (bio10), the mean temperature of the coldest quarter (bio11), the annual precipitation (bio12), the precipitation of the wettest month (bio13), the precipitation of the driest month (bio14), the precipitation seasonality (bio15), the precipitation of the wettest quarter (bio16), the precipitation of the driest quarter (bio17), the precipitation of the warmest quarter (bio18), and the precipitation of the coldest quarter (bio19). These variables are often used in species distribution modeling and could be correlated with the distribution of bryophytes.

### 2.2. Current Distribution Modeling

A thousand random background points were created in addition to the observed occurrences. The random cross-validation techniques were avoided, due to an increased underestimation of the prediction error [28]. The occurrences of *Leucobryum aduncum* and background points were separated into four equally sized blocks for spatial block cross-validation using the ENMeval package version 2.0.4 [29] in R software version 4.3.1 [30]. Then, highly correlated variables were removed using the variable selection function to avoid over-fitting the model from multicollinearity. The model hyperparameters were also tuned using the SDMtune package version 1.3.1 [31] in R software. The spatially related bioclimatic variables were removed by performing a leave-one-out jackknife test among the correlated variables, removing the variable with a Spearman's correlation coefficient greater than 0.7. Then, the variable that decreased the model performance in terms of

the area under the receiver operator curve (AUROC or AUC) was also removed. All species distribution models in this study were performed with Maxent algorithms [32] in the SDMtune package. The hyperparameter which provided the highest AUC was selected, including 1.5 regularization multiplier, linear, and quadratic transformations of input covariates. The species distribution maps and response curves of each important variable were also plotted.

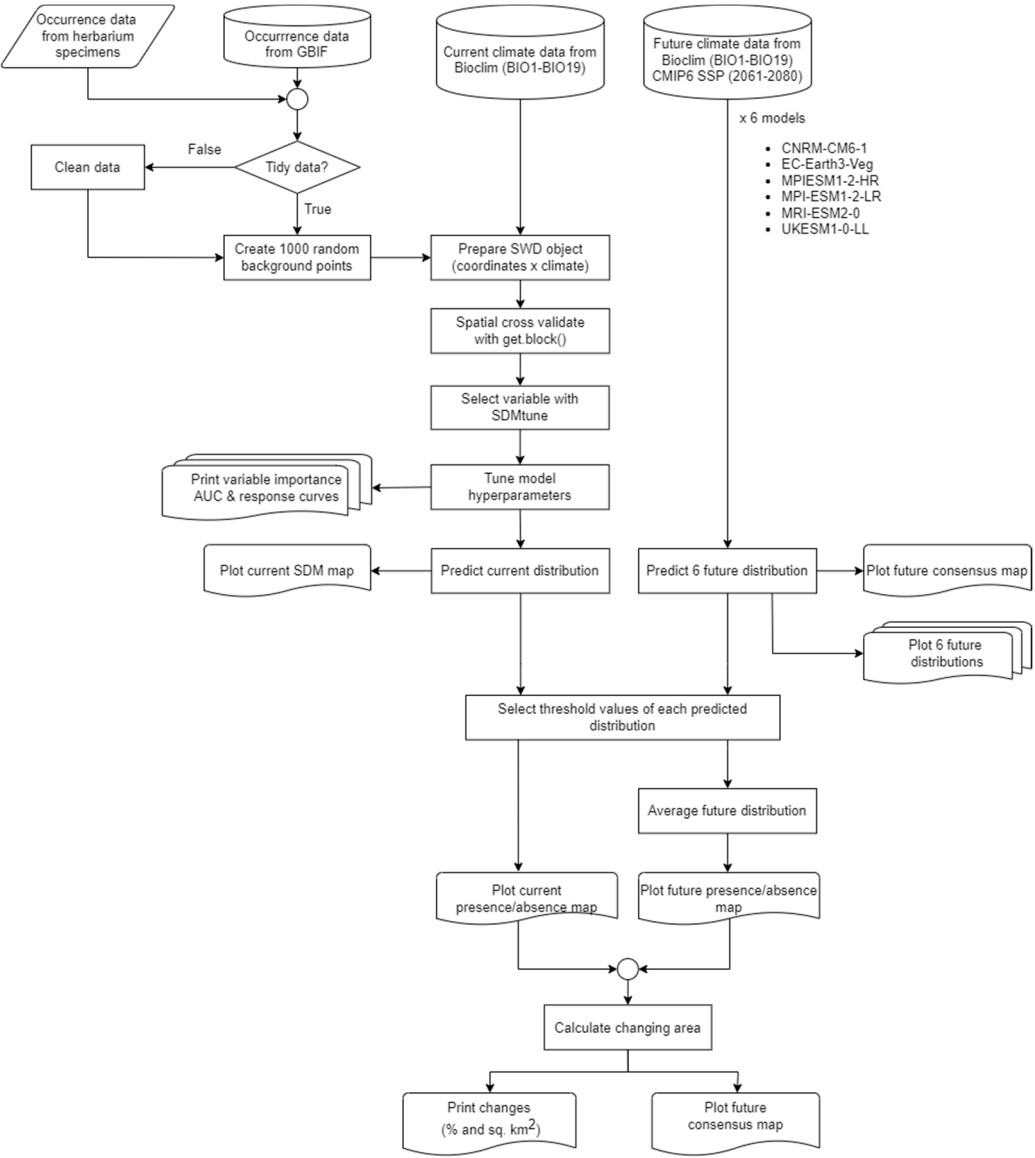

**Figure 2.** Workflow of the species distribution modeling of *Leucobryum aduncum* in the current study.

### 2.3. Predicting Future Distribution

The global climatic models (GCMs) were part of the Coupled Model Intercomparison Project Phase 6 (CMIP6) of the IPCC's sixth assessment report (AR6). Six GCMs, including CNRM-CM6-1, EC-Earth3-Veg, MPIESM1-2-HR, MPI-ESM1-2-LR, MRI-ESM2-0, and UKESM1-0-LL, from the historical period (1970 to 2000) and future period (2061 to 2080) were selected as the most suitable models for Southeast Asia according to the previous literature [33–36]. This study selected the Shared Socioeconomic Pathway (SSP2-4.5) to represent the moderate scenario for greenhouse gas emissions [37]. The current and future model predictions were performed using historical and future GCMs, the same method as in the previous section.

To calculate the change in distribution between the current and future models, we used the cloglog threshold equal to 0.57, which gives maximum training sensitivity plus specificity, to classify the area into presence and absence. Finally, the current and future distributions were compared to calculate the changes in distributions in the next 50 years. To evaluate the uncertainty among GCMs, we also created a consensus map among the six GCMs [38]. The same threshold at 0.57 was used to identify the predicted presence in each scenario.

## 3. Results

### 3.1. Selection of Climate Variables

After removing the highly related bioclimatic variables, six of the topmost important variables were selected to perform SDM. These variables included the mean diurnal range (bio2), the maximum temperature of the warmest quarter (bio5), the annual temperature range (bio7), the mean temperature of the driest quarter (bio9), the annual precipitation (bio12), and the precipitation of the warmest quarter (bio18). Among the selected variables, the two most important variables were the mean temperature of the driest quarter (bio9) and the annual temperature range (bio7), together accounting for 67.48% of permutation importance (Table 1).

**Table 1.** The chosen variables for the species distribution modeling of *Leucobryum aduncum*, sorted from most important to least important according to permutation importance.

| Abbreviation | Variable | Permutation Importance | SD |
|:---:|:---:|:---:|:---:|
| bio9 | Mean temperature of the driest quarter | 34.98 | 5.54 |
| bio7 | Annual Temperature range (bio5–bio6) | 32.50 | 4.14 |
| bio18 | Precipitation of the warmest quarter | 9.93 | 5.31 |
| bio2 | Mean diurnal range (mean of monthly (max temp.–min temp.)) | 8.08 | 3.36 |
| bio12 | Annual precipitation | 7.70 | 2.98 |
| bio5 | Max temperature of the warmest month | 6.88 | 2.63 |

The response curves of the two most important variables (the mean temperature of the driest quarter and the annual temperature range) are shown in Figure 3. The highest occurrence probability was found when the mean temperature of the driest quarter was lower than 22 °C and when the annual temperature range (the mean difference between the maximum and minimum monthly temperatures) was lower than 12 °C.

The other variables that were removed during the model selection include the mean temperature (bio1), the isothermality (bio3), the temperature seasonality (bio4), the min temperature of the coldest month (bio6), the mean temperature of the wettest quarter (bio8), the mean temperature of the warmest quarter (bio10), the mean temperature of the coldest quarter (bio11), the precipitation of the wettest month (bio13), the precipitation of the driest month (bio14), the precipitation seasonality (bio15), the precipitation of the wettest quarter (bio16), the precipitation of the driest quarter (bio17), and the precipitation of the coldest quarter (bio19).

### 3.2. Current and Future Distribution

With the Maxent algorithm, the current distribution modeling with six climate variables yielded a reasonable performance at the AUC value of 0.736. As for the future distribution, six climate scenarios, including CNRM-CM6-1, EC-Earth3-Veg, MPIESM1-2-HR, MPI-ESM1-2-LR, MRI-ESM2-0, and UKESM1-0-LL, revealed a similar future distribution (Supplementary Figures S1–S6) of *L. aduncum*. The consensus map showed that the predicted future distributions were mostly consistent across the six scenarios (Figure 4 and Table 2). The predicted current distribution included mostly the mountainous areas of Eastern Myanmar, Western Myanmar, Southern Vietnam, Northern Thailand, Upper Cen-

tral Thailand, Western Thailand, Northern Borneo, North Sumatra, South Sumatra, West Java, Bali, Sulawesi, Maluku Islands, Lesser Sundar Islands, and West Papua (Figure 5a,c). Because the six scenarios showed similar results, all six models were averaged into a single future distribution map and showed a similar distribution to the current one, but shrinking in areas in every region (Figure 5b,d).

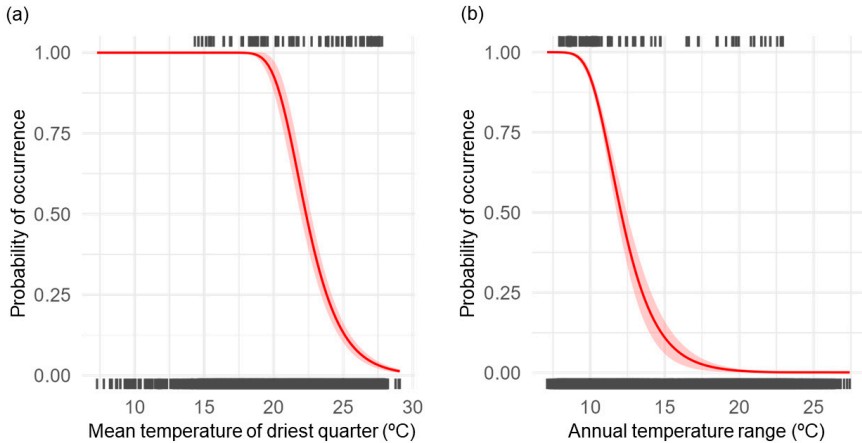

**Figure 3.** Response curves: (**a**) mean temperature of the driest quarter (bio9); (**b**) annual temperature range (bio7). The red lines represent the logistic response curves and their associate 95% confidence interval. The rug lines at 0 and 1 represent the data where the species is absent and present, respectively.

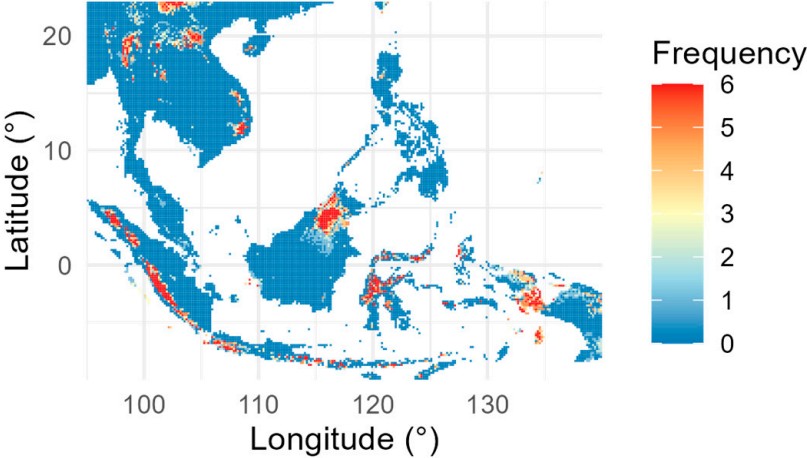

**Figure 4.** The consensus map of future habitat suitability (2061–2080), CMIP6 SSP2-4.5 from six models.

**Table 2.** The consensus area of the future habitat suitability (2061–2080), CMIP6 SSP2-4.5 from six models.

| Frequency | Area (km$^2$) | Percentage |
|:---:|:---:|:---:|
| 0 | 3,803,472.00 | 82.49 |
| 1 | 215,050.40 | 4.66 |
| 2 | 120,246.20 | 2.61 |
| 3 | 81,702.23 | 1.77 |
| 4 | 91,205.94 | 1.98 |
| 5 | 109,874.70 | 2.38 |
| 6 | 189,227.40 | 4.10 |

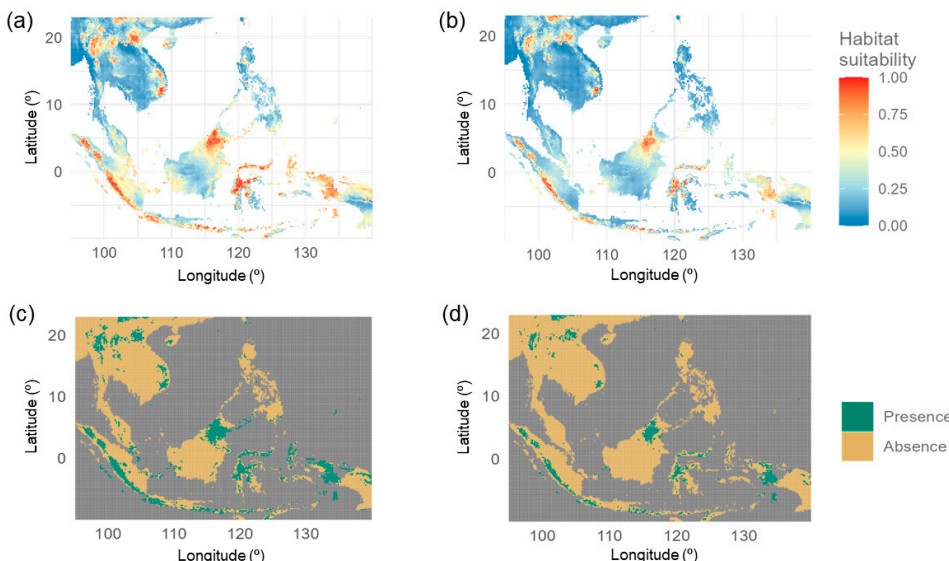

**Figure 5.** Predicted current distribution and future distribution of *Leucobryum aduncum* in Southeast Asia: (**a**) current habitat suitability; (**b**) future habitat suitability (2061–2080); (**c**) current presence–absence using the threshold = 0.57; (**d**) future presence–absence map using the threshold = 0.57.

### 3.3. Changes in Future Distribution

At the threshold of 0.57, the suitable area for *L. aduncum* presence was estimated to cover 804,957.50 km$^2$ or 17.46% of the studied area in Southeast Asia. The predicted future distribution showed that the suitable area would likely drop to 411,563.30 km$^2$, or 8.93% of the studied area, in 2061–2080 (Table 3). Most of the current distribution in the mountainous areas would shrink, except for some areas in Northern Vietnam (Figure 6).

**Table 3.** Current and future (2061–2080) areas of presence and absence for *Leucobryum aduncum*.

| Class | Current Distribution | | Future Distribution | |
|---|---|---|---|---|
| | km$^2$ | Percentage | km$^2$ | Percentage |
| Absence | 3,805,821.00 | 82.54 | 4,198,314.00 | 91.07 |
| Presence | 804,957.50 | 17.46 | 411,563.30 | 8.93 |

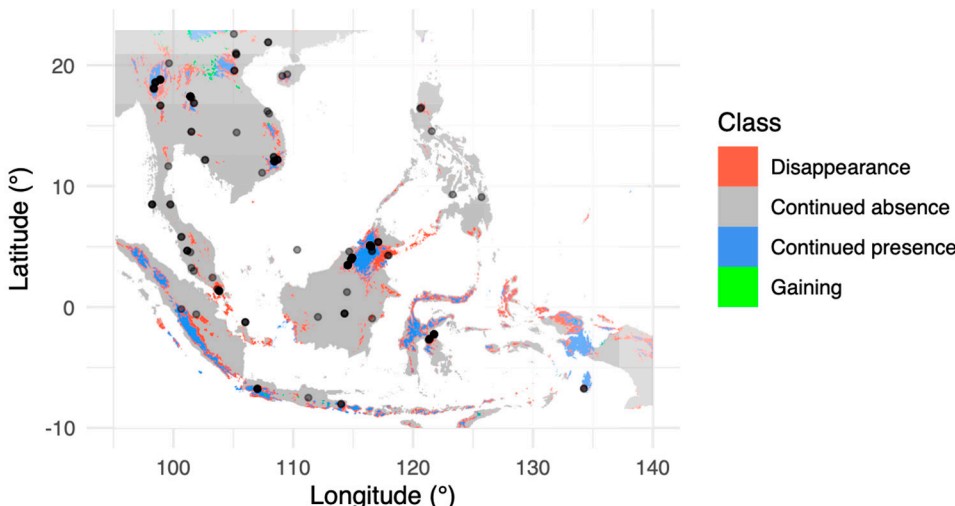

**Figure 6.** Changes in suitable areas from current to future distribution (2061–2080) for *Leucobryum aduncum*. The dots represent current occurrence locations that were used in the model.

Approximately 403,670 km$^2$, or 50% of the predicted current distribution area, would no longer be suitable for *L. aduncum* in 2061–2080. These areas were mostly at the margin of the current distribution. The core areas of the current distribution would continue to accommodate *L. aduncum*, accounting for 401,130 km$^2$ (50% of the predicted current distribution area). The areas that experience a gain in distribution are relatively small in Northern Vietnam, accounting for 10,433 km$^2$ (Table 4).

**Table 4.** Calculation of changes in the areas of the future scenario.

| Class | Area | | Percentage of Area Compared to Current Presence |
|---|---|---|---|
| | km$^2$ | Percentage | |
| disappearance | 403,669.80 | 8.76 | 50.16 |
| continued absence | 3,794,644.00 | 82.32 | - |
| continued presence | 401,130.30 | 8.70 | 49.84 |
| gaining | 10,432.99 | 0.23 | 1.30 |

## 4. Discussion

### 4.1. Predicted Distribution of L. aduncum

Species distribution modeling of *L. aduncum* showed that the mean temperature of the driest quarter and the annual temperature range were the most important variables in determining the occurrence of the species. The species showed the highest probability of occurrence at a relatively low temperature (<22 °C) and small temperature range (<12 °C). Precipitation factors played a less dominant role in determining the distribution of *L. aduncum*, suggesting that its capacity to retain additional water in hyaline cells enables it to thrive across a diverse range of water environments.

About 17% of Southeast Asia was predicted to be suitable for *L. aduncum* under the current climate conditions. Most suitable areas were limited to areas with high elevation and, subsequently, lower temperatures and smaller temperature ranges. This finding is consistent with the reported distribution of the genus *Leucobryum* in Asia [10].

The future climate in 2061–2080 indicated a warmer and generally drier climate than the current one. Subsequently, the predicted suitable areas for *L. aduncum* would be reduced by more than half compared to the current distribution. The disappearance of the present areas was mostly at the margin of the current distribution, with an exception for relatively small areas in Northern Vietnam. This shift is likely the result of an increased temperature at the lower elevation, subsequently making the distribution of this moss species even more restricted to the highlands, similar to previous studies [23,24].

### 4.2. Temperature and L. aduncum Distribution

The current study showed that the two most important variables for predicting *L. aduncum* distribution were related to temperature. For bryophytes, temperature determines the biochemical functioning and available water needed for photosynthesis. Plants will be dormant if water sources are lacking and the temperature is high. Furthermore, some research suggested some epiphytic species can be used as an indicator of changes in the microclimate. Some researchers found that microclimate values, including maximum temperature and minimum humidity, have a positive relationship with the community composition of bryophytes [39].

A decline in bryophyte richness and diversity is associated with an increased temperature by 1.5–3 °C in situ [40]. Therefore, global change may lead to the decreasing diversity of bryophytes, especially changes in species composition patterns. In long-term warming experiments in the Arctic and sub-Arctic [41], the species richness of mosses, except for *Sphagnum*, would decrease with increasing temperature. The *Sphagnum* has a higher performance to resist in higher relative temperatures compared to other northern mosses. Changes in the abundance of some bryophyte species can be bioindicators for increasing temperature. While *Leucobryum* species are superficially similar to *Sphagnum*

in their morphology, their ecological requirements are vastly different. Most *Leucobryum* species are distributed in the tropics, where the effects of increasing temperatures are less well-documented. Our study on *L. aduncum* provides additional evidence for an adverse effect of increasing temperatures on the distribution of tropical bryophytes.

Interestingly, *Leucobryum aduncum* appears to be less sensitive to precipitation. As non-vascular plants, bryophytes are poikilohydric. They are likely to require direct regulation through ambient humidity and, therefore, rely on atmospheric precipitations for water uptake. However, *Leucobryum* is distinctive in that its tissues contain empty and specialized water-holding hyaline cells, which possess the incredible ability to absorb water, exceeding several times their own dry weight. Our studied species also contains the projecting end of undulate and spinosely porated cells at the leaf apex (Figure 1c), which may help further improve their water-holding capacity [42]. The species of this genus often form a cushion, allowing a greater water-holding capacity than typical bryophytes, even in a dry environment. Our results are consistent with the previous study of *Leucobryum glaucum* in Great Britain that exhibits the highest growth rates during summer [43]. The growth of these cushions is more closely associated with temperature than with precipitation. A previous species distribution modeling study also found that higher values of annual temperature range over the year led to a lower habitat suitability for bryophytes of Orthotricaceae [44]. A temperature increase in the tropics could be more severe than the change in precipitation and could subsequently impact the distribution of tropical bryophytes.

*4.3. Other Important Variables for Bryophyte Distribution*

The diversity of bryophytes is related to many additional ecological factors [45–47], including biotic factors such as grazing and interaction with vascular plants, and physical factors such as habitat diversity, island biogeography, temperature [48], and precipitation [42]. Understanding the relationships of these factors with bryophyte diversity will help us to predict the distribution pattern of bryophytes across regional scales. It will benefit conservation and research in the global warming situation [49]. Moreover, climate change will also affect the stands and compositions of trees, which will subsequently shape microclimatic conditions in ecosystems. In the current study, we have not included the potential changes in vascular plant distribution as a factor for the future distribution of *L. aduncum* because we still have a limited understanding of how vascular plant communities affect distribution bryophytes. Future species distribution modeling of bryophytes should explore these interactions further.

**5. Conclusions**

Our study used the available climate data and current occurrences to perform the species distribution modeling of *Leucobryum aduncum*. We identified suitable areas for the species across Southeast Asia, but the prediction for the future distribution showed a dramatic decrease in suitable areas, mostly due to the increased temperature and temperature range over the next fifty years. The approach taken in the current study can be applied to other bryophyte species to understand the diverse scenarios for plant distribution and the conservation of particularly sensitive groups like tropical bryophytes.

**Supplementary Materials:** The following supporting information can be downloaded at: https://www.mdpi.com/article/10.3390/d16020125/s1, Figures S1–S6: Future 2061–2080, CMIP6 SSP2-4.5 from six models.

**Author Contributions:** Conceptualization, P.C., E.K. and N.S.; methodology, P.C. and E.K.; formal analysis, P.C.; investigation, P.C.; resources, P.T.; data curation, P.C. and P.T.; writing—original draft preparation, P.C.; writing—review and editing, E.K., P.T. and N.S.; visualization, P.C.; supervision, E.K. and N.S.; project administration, E.K.; funding acquisition, P.C., P.T. and E.K. All authors have read and agreed to the published version of the manuscript.

**Funding:** This research was funded by the scholarship to P.C. from the Development and Promotion of Science and Technology Talents Project (DPST) under the Royal Thai Government. For additional

funding for E.K. and P.T., the authors would like to thank Kasetsart University for research funding through the Biodiversity Center at Kasetsart University.

**Institutional Review Board Statement:** Not applicable.

**Data Availability Statement:** A dataset and R codes for modeling are available at https://github.com/pooh39777/sdm-leucobryum (accessed on 27 December 2023).

**Acknowledgments:** We thank Linda Beaumont from the Faculty of Science and Engineering, Macquarie University, NSW, for helping with SDM techniques and improving our models.

**Conflicts of Interest:** The authors declare no conflicts of interest.

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
