# Peer review of "Predicting the Future Distribution of Leucobryum aduncum under Climate Change"

_diversity, doi:10.3390/d16020125_

Round 1

Reviewer 1 Report

Comments and Suggestions for Authors

Comments to the Co-responding Author

This research addresses the impact of climate change on the potential distribution of a Leucobryum species. The authors model the current and future distribution of Leucobryum aduncum using the species distribution model. While I believe this kind of study is of broad interest and the manuscript has been well written, I would like the authors to address the questions and notes I wrote on the manuscript.

In the introduction section, I think you should specify the significance of selecting Leucobryum aduncum for conducting this study.

Regarding the occurrences of this species, you should explain how to avoid sampling bias.

In the discussion section, in view of the inconsistency between your results and many previous studies, I think you should focus on the explanation of why and how is temperature as an important drivers for the distributions of Leucobryum species, or bryophytes in general, as compared to water-related variables.

Comments on the Quality of English Language

The manuscript has been well written in English.

Author Response

Reviewer 1

This research addresses the impact of climate change on the potential distribution of a Leucobryum species. The authors model the current and future distribution of Leucobryum aduncum using the species distribution model. While I believe this kind of study is of broad interest and the manuscript has been well written, I would like the authors to address the questions and notes I wrote on the manuscript.

> Thank you for reviewing and providing us feedback on the manuscript. Per your suggestions, we have edited our manuscript, as indicated in the red text. We have also responded to individual comments in the text below.

In the introduction section, I think you should specify the significance of selecting Leucobryum aduncum for conducting this study. (also commented on L.48-49)

 > We have added the justification for Leucobryum aduncum in the introduction.

Regarding the occurrences of this species, you should explain how to avoid sampling bias. (also commented on L.69)

 > Due to a small sample size, we had to use all available verified data in the analysis. We reduced the risk of spatial autocorrelation by using spatial blocks in the subsequence analysis.

In the discussion section, in view of the inconsistency between your results and many previous studies, I think you should focus on the explanation of why and how is temperature as an important drivers for the distributions of Leucobryum species, or bryophytes in general, as compared to water-related variables.

The findings is inconsistent with the findings of many previous studies on the relationship between species richness of bryophytes and macroclimate factors, such as
Qian, H., & Chen, S.-B. (2016). Reinvestigation on species richness and environmental correlates of bryophytes at a regional scale in China. Journal of Plant Ecology 9: 734-741. Song, X.T., Fang, W.Z., Chi, X.L., et al., 2021. Geographic pattern of bryophyte species richness in China: The influence of environment and evolutionary history. Front. Ecol. Evol. 9, 680318.

In fact, because bryophytes, as a group of poikilohydric plants, lack vascular tissue and outer cuticle, water content of bryophytes is directly regulated by ambient humidity, and most species rely on atmospheric precipitations for water uptake, it is commonly assumed that water availability is an important driver of distributions of bryophytes, as compared to temperature-related variables (Patiño & Vanderpoorten, 2018). So, how to explain your results?

The content of this section 4.3 should focus on the explanation of why and how temperature is as an important drivers for the distributions of Leucobryum species, or bryophytes in general, as compared to water-related variables.

> Leucobryum has unique anatomical structures and growth habits that allow extra water-holding capacity, which explains why temperatures affected the distribution more than water. We have added the explanation to the discussion.

The content of this section (4.2) is a bit off topic.

> We have rewritten this section to be more connected to our results.

Reviewer 2 Report

Comments and Suggestions for Authors

The submitted manuscript is not very innovative, and research on one species poorly represented by occurrence data has been used here to formulate very speculative Conclusions. Below I give my main reservations.

1. The Bryophytes are a group of plants that is difficult to identify species, hence there are many erroneous records in the databases. Did the authors somehow verify the quality of the analyzed occurrence data?

2. The research concerns one species (Leucobryum aduncum) without rational justification why they were undertaken at all. Is it a species for some reasons particularly ecologically important or threatened by climate change?

3. The authors wrote„ The species distribution model (SDM) is a tool for predicting the potential distributions of any living organisms from the relationship between the presence/absence of the species and ecological variables.” At the same time, it would seem that they should be aware that the Maxent algorithm used in their research does not require absence data. It uses pseudo-absences. The poor description in Maxent's Materials and methods analysis suggests that they probably don't really know its basics. The estimation uncertainty maps between GCMs have not been calculated, this does not allow to assess how reliable the models are. 

4. The introduction is not very informative. Instead of focussing on the research problem, the authors are already discussing the importance of their results.

5. The study was performed using 172 occurrences, that's very little. There is no information about whether sampling bias was removed. This is common in such data, because the density of points in some locations may be greater than in others and it does not represent the actual distribution at all, and for example, differences in efforts to map biodiversity resulting from the availability of researchers, social and cultural differences of the population, etc. In addition, more detailed information about Maxent analysis parameters is missing. It looks as if there were data and the analysis was unreflectively launched, because somewhere there was a nice R script that turned out to be not entirely understandable.

6. L.129-135: This paragraph is unnecessary.

7. „Water and precipitation played a dominant role in describing the variation of species diversity, while woody species were much more strongly affected by temperature [27]”. This is not always the case. For example, most trees in temperate climates in models have a higher importance for precipitation than temperatures.

8. Chapter 4.3 is written as if a study on one species L. aduncum solved all issues related to the factors shaping the distribution of all species of mosses. 

9. It was not mentioned that climate change will also affect changes in stands, and trees strongly shape microclimatic conditions, soil, etc. in ecosystems. This should be described as a study limitation.

10. Studies on a single and poorly sampled species are not the basis for formulating conclusions about the climatic-driven distribution of all bryophytes species.

11. The engravings and tables are reasonable and legible. The text, however, requires language corrections. The biggest problem, however, is the substantive quality of the Introduction and Discussion and Conclusions. Hence, I do not recommend this manuscript for publication in the Diversity journal.

Comments on the Quality of English Language

I recommend rejecting this manuscript with the possibility of resubmission because too much should be improved. The introduction and discussion with the conclusions should be rewritten. Moreover, it seems the authors do not know what they counted and how to interpret it. Authors should perform we have uncertainty estimates between GCMs. This can be done as in these studies:

Puchałka, R., Paź-Dyderska, S., Dylewski, Ł., Czortek, P., Vitkova, M., Sadlo, J., Klisz, M., Koniakin, S., Carni, A., Rasomavicius, V., De Sanctis, M., Dyderski, M., 2023. Forest herb species with similar European geographic ranges may respond differently to climate change. Sci. Total Environ. 905, 167303. https://doi.org/10.1016/j.scitotenv.2023.167303

Author Response

Reviewer 2

The submitted manuscript is not very innovative, and research on one species poorly represented by occurrence data has been used here to formulate very speculative Conclusions. Below I give my main reservations.

> Thank you very much for your constructive feedback. We have addressed your concerns in the section below.

  1. The Bryophytes are a group of plants that is difficult to identify species, hence there are many erroneous records in the databases. Did the authors somehow verify the quality of the analyzed occurrence data?

> Leucobryum is a distinct group of bryophytes easily recognized in the field. While identifying the species can be difficult, we have tried to select the verified records from well-known herbaria, many of which contain photos and/or access to specimens for us to re-check the occurrence records.

  1. The research concerns one species (Leucobryum aduncum) without rational justification as to why they were undertaken at all. Is it a species for some reasons particularly ecologically important or threatened by climate change?

> We have added the rationale for this species in the introduction.

  1. The authors wrote„ The species distribution model (SDM) is a tool for predicting the potential distributions of any living organisms from the relationship between the presence/absence of the species and ecological variables.” At the same time, it would seem that they should be aware that the Maxent algorithm used in their research does not require absence data. It uses pseudo-absences. The poor description in Maxent's Materials and methods analysis suggests that they probably don't really know its basics. 

> We have corrected the statement about species distribution modeling. We have expanded the detailed description of the MaxEnt method in the method section as well.

The estimation uncertainty maps between GCMs have not been calculated, this does not allow to assess how reliable the models are. 

> We have added the uncertainty map to the manuscript.

  1. The introduction is not very informative. Instead of focussing on the research problem, the authors are already discussing the importance of their results.

> We have rewritten the introduction to focus more on the research problem.

  1. The study was performed using 172 occurrences, that's very little. There is no information about whether sampling bias was removed. This is common in such data, because the density of points in some locations may be greater than in others and it does not represent the actual distribution at all, and for example, differences in efforts to map biodiversity resulting from the availability of researchers, social and cultural differences of the population, etc.

> We are aware of the limitations in our dataset. The relatively small number of occurrences resulted from a rigorous screening process to ensure the specimens' identities. We reduced the risk of spatial autocorrelation by using spatial blocks in the subsequence analysis.

In addition, more detailed information about Maxent analysis parameters is missing. It looks as if there were data and the analysis was unreflectively launched, because somewhere there was a nice R script that turned out to be not entirely understandable.

> We have added the details on hyperparameter tuning and other additional details in the method section. We have developed our own analysis pipeline in R, which is available on GitHub here https://github.com/pooh39777/sdm-leucobryum.

  1. L.129-135: This paragraph is unnecessary.

> We have rewritten the section to better reflect our results.

  1. „Water and precipitation played a dominant role in describing the variation of species diversity, while woody species were much more strongly affected by temperature [27]”. This is not always the case. For example, most trees in temperate climates in models have a higher importance for precipitation than temperatures.

> We have corrected the statement to cover only species from Southeast Asia as suggested from [27].

  1. Chapter 4.3 is written as if a study on one species L. aduncum solved all issues related to the factors shaping the distribution of all species of mosses. 

> We have rewritten the section to be more specific to L. aduncum.

  1. It was not mentioned that climate change will also affect changes in stands, and trees strongly shape microclimatic conditions, soil, etc. in ecosystems. This should be described as a study limitation.

> We have described this in the discussion.

  1. Studies on a single and poorly sampled species are not the basis for formulating conclusions about the climatic-driven distribution of all bryophytes species.

> We apologize if we have given that impression for the reader. We are hoping that our methodology can be used to study other bryophytes in the future. We have adjusted the conclusions accordingly.

  1. The engravings and tables are reasonable and legible. The text, however, requires language corrections. The biggest problem, however, is the substantive quality of the Introduction and Discussion and Conclusions. Hence, I do not recommend this manuscript for publication in the Diversity journal.

> We have improved upon our introduction and discussion which will hopefully be now worthy of publication in Diversity.

Comments on the Quality of English Language

I recommend rejecting this manuscript with the possibility of resubmission because too much should be improved. The introduction and discussion with the conclusions should be rewritten. Moreover, it seems the authors do not know what they counted and how to interpret it. Authors should perform we have uncertainty estimates between GCMs. This can be done as in these studies:

Puchałka, R., Paź-Dyderska, S., Dylewski, Ł., Czortek, P., Vitkova, M., Sadlo, J., Klisz, M., Koniakin, S., Carni, A., Rasomavicius, V., De Sanctis, M., Dyderski, M., 2023. Forest herb species with similar European geographic ranges may respond differently to climate change. Sci. Total Environ. 905, 167303. https://doi.org/10.1016/j.scitotenv.2023.167303

> We have improved upon our introduction and discussion. We have also added the consensus map to address the uncertainty as found in the suggested reference. We have included the maps of all scenarios in the supplementary materials.

Reviewer 3 Report

Comments and Suggestions for Authors

Dear Authors,

Congratulation the interesting results. As I can see, in the submitted manuscript, the available data on the distribution of the Leucobryum aduncum species and subsequently also the climate data were used. According to your results, ongoing climate change may change the area of current occurrence. Both the current areas suitable for the occurrence of the species, as well as a forecast of future distribution is presented. This shows a dramatic decline in such suitable sites. The approach used in this study has been used in the past and is shown to work quite well for bryophytes. It's a pity that you cite so few articles with a similar focus, because it could give weight to your results - it would show the weight of such studies and especially when using bryophytes as an object of study. I especially recommend Lukas Cihal's articles - for creating a strong discussion and confirming the methods used. Otherwise, I rate the manuscript as successful and I see the use of the results especially in nature conservation.

Author Response

Dear Authors,

Congratulation the interesting results. As I can see, in the submitted manuscript, the available data on the distribution of the Leucobryum aduncum species and subsequently also the climate data were used. According to your results, ongoing climate change may change the area of current occurrence. Both the current areas suitable for the occurrence of the species, as well as a forecast of future distribution is presented. This shows a dramatic decline in such suitable sites. The approach used in this study has been used in the past and is shown to work quite well for bryophytes. It's a pity that you cite so few articles with a similar focus, because it could give weight to your results - it would show the weight of such studies and especially when using bryophytes as an object of study. I especially recommend Lukas Cihal's articles - for creating a strong discussion and confirming the methods used. Otherwise, I rate the manuscript as successful and I see the use of the results especially in nature conservation.

> Thank you very much for your positive feedback. We have added more citations that hopefully increase more evidence for our study.

Round 2

Reviewer 1 Report

Comments and Suggestions for Authors

Although the relevant issues and suggestions have been adopted and significant revisions and improvements have been made to the manuscript, I still suggest that the authors should make necessary improvements for the introduction to better highlight the research topic, rather than providing some broader introductions for bryophytes. In addition, the authors should carefully check the manuscript to ensure the accuracy of the expressed meaning. For example, in line 61-62, the sentence should be expressed with the opposite meaning.

Author Response

Reviewer 1

Although the relevant issues and suggestions have been adopted and significant revisions and improvements have been made to the manuscript, I still suggest that the authors should make necessary improvements for the introduction to better highlight the research topic, rather than providing some broader introductions for bryophytes.

  • We have added more context for our studied species in the introduction, including photos of plant habits and anatomical structure, to better highlight the research topic. The additional content is highlighted in red.

In addition, the authors should carefully check the manuscript to ensure the accuracy of the expressed meaning. For example, in line 61-62, the sentence should be expressed with the opposite meaning.

  • We have checked the manuscript for accuracy and made necessary changes as highlighted in red text.

Reviewer 2 Report

Comments and Suggestions for Authors

The authors have followed most of my comments, but I have two more that they should consider.

1. L138-140: To evaluate the uncertainty among GCMs, we also created a consensus map among the six GCMs. The same threshold at 0.57 was used to identify the predicted presence in each scenario.  - The authors of the method of estimating the uncertainty among GCMs should support by quoting the literature.

2. Still the weak side of this manuscript is the lack of a good connection between bioclimatic variables and the importance for biology and the occurrence of the species under study. It is worth trying, correcting this work so that it does not look like many other ad-hoc written papers. This should increase of scientific soundness of this paper.

Kind regards

Author Response

Reviewer 2

The authors have followed most of my comments, but I have two more that they should consider.

  1. L138-140: To evaluate the uncertainty among GCMs, we also created a consensus map among the six GCMs. The same threshold at 0.57 was used to identify the predicted presence in each scenario.  - The authors of the method of estimating the uncertainty among GCMs should support by quoting the literature.
  • We have added the related references for evaluating uncertainty using the consensus method.

  1. Still the weak side of this manuscript is the lack of a good connection between bioclimatic variables and the importance for biology and the occurrence of the species under study. It is worth trying, correcting this work so that it does not look like many other ad-hoc written papers. This should increase of scientific soundness of this paper.
  • We have added the discussion on the connection between bioclimatic variables and the importance for biology and the occurrence of Leucobryum aduncum, as well as the additional images for the habit and anatomical structure of plants. We hope that the addition helps the reader understand more about the results and biology of the species.

Round 3

Reviewer 2 Report

Comments and Suggestions for Authors

I don't think the authors will be able to improve anything else here.